# MetS Prevalence and Its Association with Dietary Patterns among Chinese Middle-Aged and Elderly Population: Results from a National Cross-Sectional Study

**DOI:** 10.3390/nu14245301

**Published:** 2022-12-13

**Authors:** Pengkun Song, Xiaona Zhang, Yuqian Li, Qingqing Man, Shanshan Jia, Jian Zhang, Gangqiang Ding

**Affiliations:** 1National Institute for Nutrition and Health, Chinese Center for Disease Control and Prevention, 29 Nanwei Road, Xicheng, Beijing 100050, China; 2Key Laboratory of Trace Elements and Nutrition of National Health Commission, Beijing 100050, China

**Keywords:** metabolic syndrome, dietary pattern, cross-sectional study, China

## Abstract

The prevalence of metabolic syndrome (MetS) increased dramatically over the past years among adults in a separate province in China; little is known about newly diagnosed MetS in middle-aged and above residents nationwide. We investigated the prevalence of MetS and its components, the dietary patterns, and their relationship among the middle-aged and above population of China by using data from a national cross-sectional survey. General information involving lifestyles and health stations was collected, and dietary intake using a 3-day 24 h dietary recall and weighing method for edible oil and condiments was conducted. Height, weight, waist circumference, and blood pressure were measured, and fasting serum lipids and glucose were tested by trained clinical staff. Dietary patterns were derived from 23 food categories by using cluster analysis, and a multivariate logistic regression model was used to evaluate the odd ratio of MetS and its component across obtained dietary patterns. The estimated prevalence of MetS was 37.1% among 40,909 middle-aged and older participants in the study. Participants were classified into diversity pattern, northern pattern, and southern pattern that, respectively, accounted for 9.8%, 47.2%, and 43.0% of the total. Compared with those inclined to the northern pattern, participants prone to the southern pattern decreased the risk of MetS (OR = 0.81, 95%CI: 0.75- 0.87; *p* < 0.001), central obesity (OR = 0.70, 95%CI: 0.65–0.76; *p* < 0.001), and HDL-C (OR = 0.82, 95%CI: 0.76–0.89; *p* < 0.001), and elevated BP (OR = 0.86, 95%CI: 0.79–0.93; *p* < 0.001) respectively. However, participants of the southern pattern tended to have a higher risk of elevated glucose; the OR (95%CI) was 1.13 (1.05, 1.22; *p* = 0.002) after adjusting for potential confounding factors. Greater adherence to diverse dietary patterns was negatively related to the risk of central obesity and elevated blood pressure with an OR (95%CI) of 0.82 (0.71, 0.94; *p* = 0.005) and 0.77 (0.67, 0.88; *p* < 0.001), respectively. We concluded that dietary improvement and health promotion for MetS should be based on the district-specific nutritional status of the Chinese middle-aged and elderly population.

## 1. Introduction

Metabolic syndrome (MetS) is a multiple metabolic disorder characterized by insulin resistance and mainly manifested by obesity, especially central obesity, hyperglycemia, hypertension, and dyslipidemia combination in one individual [1]. Generally, despite different definitions being used in various populations involving ethnicity, gender, and age, the prevalence of MetS is alarmingly high all around the world. According to the International Diabetes Federation (IDF) definition of MetS, one-quarter of the world’s adults were diagnosed with MetS. The prevalence of MetS in American adults increased from 27.9% in 1999 to 31.5% in 2014 [2]. With the rapid economic development and the deepening of an aging society, the prevalence of MetS in Chinese residents also undergo an increasing trend. Studies have shown that the prevalence of MetS among the elderly in China reached 36.9% in 2013 [3]. As an important public health problem, metabolic syndrome contributes to the development of diabetes mellitus, cardiovascular diseases, and premature mortality [4]. Previous studies have demonstrated that the prevalence of MetS was of a great difference in different geographic locations of China, combined with the particularity of culture, economy, living habits, and degree of aging [5,6]. Therefore, evaluating the distribution of MetS prevalence and conducting an in-depth study of the influencing factors related to MetS was of great significance for the implementation of public health measurements.

Most studies demonstrated that the development of MetS is closely related to excessive energy intake, physical inactivity, and unhealthy lifestyles [7]. Mediterranean diet, DASH diet, and MIND diet were representative of healthy dietary patterns and had been reported to be beneficial for reducing the risk of MetS comprehensively [7,8,9,10]. Diverse dietary habits in different regions of China, to some extent, caused the disparate prevalence of MetS and its components. In the present study, we intended to (1) obtain the national prevalence of MetS and main dietary patterns among the middle-aged and above population of China; (2) explore the association between dietary patterns and MetS. Thus, it may supply moderate scientific evidence of dietary nutrition for promoting the health of the elderly in China. 

## 2. Data Source and Survey Population

Data were from a nationwide cross-sectional study named China Adults Chronic Diseases and Nutrition Surveillance in 2015 (CACDNS 2015). This study was conducted in 31 provinces, autonomous regions, and municipalities throughout mainland China (except Taiwan, Hong Kong, and Macao). Multi-stage stratified cluster random sampling method was used to collect the participants aged 18y and older. The detailed sampling procedure was described [11]. Basic information, lifestyle, diet, and health status of the middle-aged and above through household and personal questionnaires, body measurements, dietary surveys, and laboratory tests. Participants aged 45 years and above were included in our study. For those with missing values of basic demographic information, samples with missing or illogical values were eliminated. Those with abnormal daily calorie intake (<800 kcal or >4000 kcal in males and <500 kcal or >3500 kcal in females) also were deleted.

### 2.1. General Information Collection and Dietary Nutrients Assessment 

A standard questionnaire was used to collect general demographic information, lifestyle, health status, and physical activity information through face-to-face interviews at residents’ homes by trained investigators. Meanwhile, we used 24 h dietary recalls for 3 consecutive days to collect dietary information. All the food intake, breakfast, lunch, dinner, as well as snack times, were collected during the period of the interview. In addition, a food-weighting record was used to collect the consumption of edible oil and main condiments such as salt, sauce, and other flavorings during the three consecutive days. All the foods were firstly coded according to the “Chinese Food Composition Table” 2018 edition, then they were classified into 23 food categories based on similar types of foods and nutrient contents (see Appendix A). Daily energy, fat, protein, and carbohydrate intake, as well as minerals and vitamin intake per day, were also evaluated. 

### 2.2. Dietary Pattern Assessment

In the study, fast cluster analysis was used to derive the main dietary patterns. Due to the cluster analysis being sensitive to outliers, we eliminated the participants whose food intakes were higher or lower than the five standard deviations of means first. Then all the food groups were standardized, and the K-means clustering algorithm was used for each food group. After continuous iteration conduction, all the participants were classified into mutually exclusive dietary pattern subgroups according to the Euclidean of each individual and cluster distance. We prespecified two to four clustered groups and the final selection of dietary groups was determined by a large variance ratio and the even distribution of the study subjects (the subjects were more than 1000 per dietary pattern subgroup). Considering some previous study suggestions [12], the number of dietary pattern classifications was three through cluster analysis combining professional knowledge. In addition, the three dietary patterns were named after traditional foods in the geographic location of China.

### 2.3. Anthropometric Measurement and Biomarkers Test

Height and body weight were measured by trained investigators in the morning when participants were in a fasting state. BMI was calculated and then divided into four categorical levels. Waist circumference was also measured in the fasting state by using a non-retractable material flexible ruler. An electronic sphygmomanometer of Omron HBP 1300 was used to determine blood pressure for three consecutive standard measurements, and then the average value was calculated for each participant with an accuracy of 0.1. 

Fasting plasma glucose and lipids were tested by a 7600 automatic biochemical analyzer (Hitachi, Japan), the detection kit provided by China Biocontrol Biotechnology Co., Ltd. Fasting plasma glucose concentration was measured by hexokinase method, triglyceride (TG) was measured by enzyme colorimetric method, and high-density lipoprotein cholesterol (HDL-C) was measured by direct elimination method with strict quality of laboratory control. 

### 2.4. Metabolic Syndrome Diagnosis 

The diagnosis of metabolic syndrome was determined according to the revised National Cholesterol Education Programme Expert Panel on Detection, Evaluation, and Treatment of High Blood Cholesterol in Adults (NCEP ATP III) criteria for Asian Americans [13]: WC ≥ 90 cm in men and ≥80 cm in women was considered to be central obesity, elevated triglyceride was determined as higher than 1.7 mmol/L or under treatment, decreased HDL-C was the level of HDL-C < 1.0 mmol/L in men and <1.3 mmol/L in women or under treatment, the elevated blood pressure was calculated with systolic blood pressure higher than 130 mmHg or diastolic blood pressure higher than 85 mmHg or being treated, and the last indicator was elevated plasma glucose of higher than 5.6 mmol/L or diagnosed to be diabetes in former. Participants with ≥3 risk factors were defined as MetS.

### 2.5. Covariant Index

In the study, the covariant variables were as follows: Age was classified into three subgroups (45–59 y, 60–74 y, 75 y-), education level (low, medium, high), living district (urban/rural), marital status (having a partner or other), nationality (Han/other), smoking (yes or no), drinking (yes or no), adequate physical activity (yes or no), annual household income (<5000, 5000-<10,000, 10,000-, no response), BMI (underweight was BMI < 18.5 kg/m^2^, normal weight was BMI 18.5–23.9 kg/m^2^, overweight was BMI 24.0–27.9 kg/m^2^, and obesity was BMI ≥ 28.0 kg/m^2^), family history of chronic diseases (yes/no), and total energy intake (continuous).

### 2.6. Statistical Analyses

SAS 9.4 version was used for data analysis (SAS Institute, Cary, NC, USA). The calculation of mean and rate was performed with complex sampling weighting. The basic sampling weights were calculated according to the sampling design, using the 2010 China 6th census data as the standard population, and the expost stratification weights were calculated by urban and rural areas, age, and gender, and the final weight was the product of the basic sampling weight and the expost stratification weight. The mean was calculated with PROC SURVEYMEANS, and the difference test was calculated with PROC SURVEYREG. Rates were calculated using PROC SURVEYFREQ. The relationship between dietary patterns and MetS and its components was analyzed by multivariate weighted logistic regression and adjusted by potential confounding covariant variables.

## 3. Results

Table 1 shows the general information and life style of the participants.There were a total of 40,909 participants (16,901 in urban and 24,008 in rural) aged 45 y and above enrolled in the survey, and the mean age was 60.4 y in males and 59.0 y in females. In urban areas, females accounted for 53.3%, and 55.4% in rural. As for education level, most participants, especially females, only had primary school education or even lower; 64.9% of the females had junior middle school and above; in males, the figure is just 2.2% in rural areas. Most of the participants were in the status of having a partner, which accounted for 91.1% for females and 90.9% for males. The percentage of smoking and drinking in males was 52.3% and 36.5% while 3.4% and 5.3% in females, respectively, the difference in the smoking and drinking rates between genders were all statistically significant (*p* < 0.001). Moreover, the adequate physical activity rate was 66.1% among males, which was significantly lower than that in females. There was a statistically significant higher BMI value and overweight and obesity percentage among females than that in male participants (*p* < 0.001).

Table 2 shows the distribution of relevant indicators of MetS, the weighted prevalence of MetS, and its five components’ prevalence among 45 y and above participants in the survey. The weighted mean of waist circumference was 83.7 cm (95%CI: 83.2 cm to 84.1 cm), the mean of SBP and DBP was 138.6 mmHg (95%CI: 137.9–139.4) and 80.3 mmHg (95%CI: 79.9–80.7) respectively. The weighted means of serum TG, HDL-C, and glucose concentration were 1.50 mmol/L, 1.28 mmol/L, and 5.50 mmol/L among the samples, respectively. The TG and glucose concentration was much higher in urban than that in rural, while the HDL-C level among urban participants was much lower than that in rural participants.

Among the middle-aged and above participants in China, the weighted prevalence of MetS was 37.1%, 43.4% in urban, 34.2% in rural, 29.5% in males, and 47.5% in females. The weighted prevalence of its components was respectively 45.4% for central obesity, 29.4% for elevated triglyceride, 39.5% for decreased HDL-C, 68.4% for elevated blood pressure, and 33.3% for elevated glucose. The higher prevalence was seen in urban areas than that in rural (central obesity: 51.3% vs. 40.2%, elevated TG: 31.9% vs. 27.3%, decreased HDL-C: 43.2% vs. 36.4%, elevated glucose: 36.3% vs. 30.7%), but a similar prevalence of elevated blood pressure was seen in urban and rural (68.2% vs. 68.6%). The prevalence of three to five components of MetS was 20.4%, 13.1%, and 5.0%, respectively, among the participants. When compared with rural participants, the value of urban participants with three to five items of MetS was also much higher, which was 22.4% vs. 18.6%, 14.5% vs. 11.9%, and 6.5% vs. 3.7%, respectively. However, in male participants, the value of the three and above components of MetS were 17.1%, 9.4%, and 3.0%, while much higher values of 23.6, 16.9%, and 7.0%, respectively, in females.

As for dietary patterns, three dietary patterns were obtained by cluster analysis, and they were named diversity pattern, northern pattern, and southern pattern, characterized by typical food consumption (Table 3). Those were classified into diversity pattern, northern pattern, and southern pattern and accounted for 9.8%, 47.2%, and 43.0% of the total participants. In the diversity pattern, participants consumed moderate rice and wheat; the quantity was 78.6 g/d and 130.1 g/d, respectively. While in participants of the northern pattern, the consumption of wheat products was 195.3 g/d and rice products was 46.7 g/d, which were relatively more wheat and less rice than that in the diversity pattern. Moreover, 192.1 g/d of rice intake and 41.6 g/d of wheat intake were seen in participants classified as southern pattern, with more rice and less wheat consumption in this pattern when compared with that in the diversity pattern. In addition, participants consumed the highest of dry legumes, legume products, light-colored vegetables, fungi and algae, fresh fruits and nuts, other red meats, dairy products, eggs, cakes and desserts, sugar, and starch in the diversity pattern. Higher intakes of other cereals, tubers, and salt were characterized in the northern pattern, while the highest consumption of dark color vegetables, salted vegetables, pork, red meat offal, poultry, aquatic products, vegetable oil, and animal oil was in the southern pattern.

Table 4 describes energy and nutrient intake in three dietary patterns. The highest daily energy intake of 1953.3 kcal/d in the diversity pattern and the lowest of 1649.9 kcal/d in the northern pattern are in Table 4. The carbohydrate, protein, dietary fiber, and cholesterol intake were the highest in diversity pattern; the value was 245.9 g/d, 67.3 g/d, 13.4 g/d, and 200.2 mg/d, respectively. There was the highest intake of vitamins of thiamin, riboflavin, vitamin A, vitamin C, and total α-vitamin E, as well as minerals intake such as calcium, copper, iron, iodine, potassium, magnesium, manganese, phosphate, selenium, and zinc in diversity pattern when compared with other two patterns. In the southern pattern, the intake of carbohydrates, dietary fiber, copper, and iodine was 221.5 g/d, 8.7 g/d, 1.5 mg/d, and 5.9 μg/d, which was the lowest level among the three patterns. Meanwhile, the highest fat intake of 83.1 g/d in the southern pattern is shown in the table. More interestingly, participants consumed the highest sodium of 7509.5 mg/d but the lowest of most nutrients except for carbohydrates, dietary fiber, copper, and iodine in the northern pattern. As for the structure of daily fat composition, much higher SFA and MUFA were consumed and relatively higher energy contribution of them in participants of the southern pattern, while the PUFA was the highest with a mean intake of 24.4 g/d in the diversity pattern.

Figure 1 demonstrates the relationships among MetS and its components and dietary patterns. Compared with participants of the northern pattern, those with the southern pattern tended to have a lower risk of MetS; the OR and 95%CI was 0.81 (0.75–0.87, *p* < 0.001) after being adjusting for age, gender, district, nationality, marital status, income, education level, family history of chronic diseases, smoking, drinking, physical activity, BMI and total energy. In terms of the components of MetS, the OR (95%CI) of central obesity, decreased HDL-C, and elevated BP for the southern pattern was 0.70 (0.65–0.76, *p* < 0.001), 0.82 (0.76–0.89, *p* < 0.001) and 0.86 (0.79–0.93, *p* < 0.001) respectively after being adjusted potential confounders. However, participants of the southern pattern tended to have a higher risk of elevated glucose; the OR (95%CI) was 1.13 (1.05, 1.22; *p* = 0.002).

There was not a significant association in MetS except for central obesity and elevated blood pressure in participants of diversity pattern when compared with those of northern pattern. The OR of MetS was 0.91 (95%CI: 0.80–1.03, *p* = 0.144) in participants with diversity patterns by adjusting confounders. Meanwhile, the ORs and 95%CI of elevated TG decreased HDL-C, and elevated glucose was 1.01 (0.88–1.16, *p* = 0.917), 1.08 (0.95–1.23, *p* = 0.257), and 0.96 (0.85–1.10, *p* = 0.59) respectively in diversity pattern. Nevertheless, there was a significant inverse relationship of diversity pattern with central obesity and elevated blood pressure, with an OR (95%CI) of 0.82 (0.71, 0.94; *p* = 0.005) and 0.77 (0.67, 0.88; *p* < 0.001) respectively for diversity pattern after being adjusted potential confounders (Figure 1).

## 4. Discussion

The present study provided an updated estimated nationwide prevalence of MetS and its association with dietary patterns among Chinese milled-aged and elders. The prevalence of metabolic syndrome in this study is relatively high, especially in urban female participants. Three dietary patterns were identified, which were named the diversity pattern, northern pattern, and southern pattern, by the characteristics of food consumption. Compared with participants intended for the northern pattern, the southern pattern was significantly inversely related to the risk of MetS, while the diversity pattern was inversely associated with some of its components after adjusting for confounding covariates.

Generally, the prevalence of MetS in adults below 50y was slightly higher in men, but it reverses after 50 years because of a change in genetical and biological pathways occurring after menopause, and females are more prone than males to develop MS in response to socio-economic status [14]. In our study, the total weighted prevalence of MetS was 37.1%, which was slightly higher than the prevalence of MetS of 34.77% in the study of CHARLS in 2015 [15] and much higher than the prevalence of 26.9% in the CSPP survey among 40y and above participants conducted in 30 provinces of China in 2014–2015 [16]. Although the same criteria were used to diagnose MetS, the discrepant results of prevalence may be attributable to the sampling procedure, different data collection methods, and the age of participants presumably. Nevertheless, the subgroup analysis in our study showed consistent results with previous studies [17,18,19]; that is, the prevalence of MetS in the urban area and females was much higher than that in rural and males.

In our study, participants for the northern dietary pattern accounted for the highest proportion of 47.2%, while those prone to the southern pattern and diversity pattern were 43.0% and 9.8%, respectively. In fact, residents living in northern areas accounted for almost 40% of mainland China, the typical diet in the north is characterized as a higher intake of wheat products and tuber, fewer vegetables, and fruits accompanied by higher edible oil and sodium. Meanwhile, the main food source of protein was eggs, pork, and poultry, and salted vegetables were also popular in the district [20]. From the chronic diseases map of China, we can see the prevalence of hypertension and CHD was much higher in this district than in others, which will be partly attributed to its dietary structure.

The majority of studies confirm quite consistently that healthy and rational dietary patterns such as the Mediterranean diet and DASH are beneficial for reducing the risk of hypertension, overweight and obesity, diabetes, and cardiovascular and cerebrovascular diseases [10,21,22,23], while patterns identified as unhealthy such as Western diet increase the risk [24,25]. In the present study, those who were adhering to the southern pattern in their diet showed a decreased risk of MetS, which demonstrated that this pattern was seemingly beneficial for MetS prevention. In fact, in the south of China, especially in the southeast coastal areas, the incidence of cardiovascular and metabolic disease was less than that in the north region [26,27]. Several articles have shown that southern patterns of China have been associated with a lower risk of MetS. With its traditional foods of rice, vegetables, fruits, and aquatic products, the southern dietary pattern is characterized by high dietary fiber and polyunsaturated fatty acids and is rich in vitamins and minerals but less saturated fat and sodium; it is beneficial to the control of blood pressure and lipid [28,29]. Shi et al. [30] used the Jiangsu Nutrition Survey to analyze the correlation between rice intake, body weight, and MetS and found that rice intake was negatively correlated with body weight and did not increase the risk of MetS. Meta-analysis showed that vegetable and fruit intake in the Asian population could reduce the risk of metabolic syndrome by 14% on average [31]. Interestingly, our study revealed that participants who were higher adherence to the southern pattern might be more likely to be at higher risk of glucose elevation. The association was weaker, but it was statistically significant after adjusting for potential confounders. In the southern patterns of our study, higher pork and less milk consumption were seen compared with the other two patterns. Several studies revealed that high consumption of red meat was positively associated with the risk of diabetes [32]. Jannasch et al. [33] evaluated 25 prospective studies involving 390,664 participants in five continents and revealed that dietary patterns characterized by higher intake of red meat and processed meat were positively associated with a higher incidence of type 2 diabetes. Another meta-analysis conducted on 50,345 Caucasians also demonstrated that unprocessed red meat consumption was related to higher glucose and insulin concentration; the concentration increased with an average of 0.037 mmol/L and 0.049 ln-pmol/L respectively, when adding 100 g/d of red meat [34]. Nevertheless, studies on the association between diabetes and dairy consumption showed some inverse results. One study conducted on the Asian population revealed that a 200 g per day of dairy product increase was negatively related to the risk of diabetes; the RR and 95%CI was 0.97 (0.94–0.99) [35]. A cohort study also demonstrated will the risk of diabetes was inversed when red meat was replaced with fish or poultry [36]. On the other hand, from the result of fatty acid components in the southern pattern, we also suggest that replacing some of the red meat with fish, poultry, and dairy products, which contain higher PUFA and lower SFA, might be a benefit for glucose concentration among those that adhere to southern pattern [37].

As for the diversity pattern in the study, the characteristic foods of this pattern were the highest consumption of dry legume and legume products, fresh vegetables and fruits, milk and dairy products, nuts, and eggs, as well as other red meat, cakes, and desserts. This pattern contained the highest intake levels of energy, carbohydrate, protein, ascorbic acid, calcium, and potassium but the lowest level of sodium intake among the three patterns. It was similar to the prudent diet reported in other countries to some extent that was inversely related to the risk of MetS [38,39]. However, in our study, we only obtained a significant inverse relationship between this pattern and central obesity and elevated blood pressure; this phenomenon has been partly explained by healthy foods, such as legumes, vegetables, fruits, dairy products, and nuts, which were a benefit for the control of central obesity and blood pressure owing to their containing more dietary fiber, abundant vitamins, and higher potassium but less sodium [40,41]. Although there was no significant association between this pattern with MetS and its components of elevated TG, decreased HDL-C, and elevated glucose when compared with the northern pattern, the diversity pattern might be improved by replacing red meat with aquatic foods concerning blood lipids profile. On the basis of the literature review, it is possible that higher fish intake is independently related to a lower risk of MetS [42,43,44].

The study also has some limitations. Firstly, the causal connection between dietary patterns and MetS developments remains uncertain because of the cross-sectional study. When some participants were diagnosed to be in the metabolic disorder station, they may possibly make some changes in diet structure or other living styles. Moreover, the three dietary patterns we derived by k-means cluster analysis, which is based on distance measures between observations of individuals, may not thoroughly explain all Chinese dietary patterns, such as some special diets in minority areas of China. Another limitation was that the dietary intake was assessed by 24 h dietary recall over 3 consecutive days in this study; dietary intake may change with time while these changes may influence the relationship between diet and MetS. Furthermore, differences in the criteria for judging metabolic syndrome and characteristic foods in each dietary pattern from other studies will inevitably lead to some certain limitations for extrapolating the relationship of dietary patterns with metabolic syndrome. Nevertheless, our study was on the basis of large and ethnically homogenous participants; those respondents who had changed dietary structure because of diagnosed chronic diseases in former were excluded; dietary patterns we used determined a posteriori, which was closer to the reality, much better than the patterns derived by previously adopted assumptions. Given the complex effect of food and nutrient consumption on health, as well as no consensus on dietary recommendations for metabolic syndrome, further cohort studies are essential for investigating the relationship between metabolic syndrome and dietary patterns.

## 5. Conclusions

In the present study, MetS and its components’ prevalence were invested, three dietary patterns among middle-aged and elderly people in China were obtained, and the association between these dietary patterns with MetS was evaluated. The study demonstrated that the three dietary patterns have their own nutritional characteristics and need to be improved for the aspect of MetS prevention and control. When compared with those inclined to the northern pattern, participants who adhered to the southern pattern were at a lower risk of MetS. In contrast, those inclined to the diversity pattern had a decreased risk of central obesity as well as blood pressure. The findings in the present study provide additional evidence of Chinese dietary patterns, and the effort of dietary quality improvement should be comprehensively based on the components heterogeneity of metabolic disorders.

## Figures and Tables

**Figure 1 nutrients-14-05301-f001:**
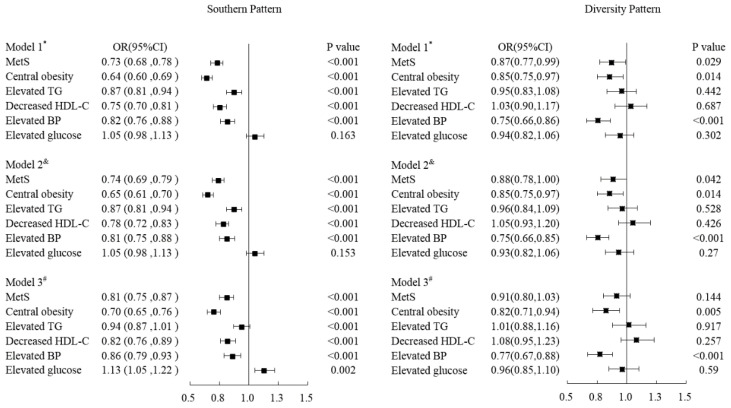
Odd ratios and 95%CI of MetS and its components of Sothern Pattern and Diversity pattern respectively comparing with the Northern Pattern by adjusting covariant variables. * Model 1: Adjusted age, gender, district, nationality, marital status, income, education level and family history of chronic diseases. ^&^ Model 2: Model 1 adding life styles such as smoking, drinking and physical activity. ^#^ Model 3: Model 2 adding the variables of BMI and energy.

**Table 1 nutrients-14-05301-t001:** General information and life style of the participants aged 45 y and above in the survey.

	Urban Area		Rural Area		Total	
	Males (*N* = 7889)	Females (*N* = 9012)	Males (*N* = 11,824)	Females (*N* = 12,184)	Males (*N* = 19,713)	Females (*N* = 21,196)
Age (mean, std, y)	61.1 (9.7)	59.9 (9.3) *	60.0 (9.3)	58.3 (8.9) *	60.4 (9.5)	59.0 (9.1) *
Age group (*n*, %)						
45–59 y	3673 (46.6)	4675 (51.9) *	6078 (51.4)	7167(58.8) *	9751 (49.5)	11,842(55.9) *
60–74 y	3486 (44.2)	3727 (41.4)	4931 (41.7)	4484 (36.8)	8417 (42.7)	8211 (38.7)
75 y	730 (9.3)	610 (6.8)	815 (6.9)	533 (4.4)	1545 (7.8)	1143 (5.4)
Han Nationality (*n*, %)	7467 (94.7)	8506 (94.4)	10,451 (88.4)	10,806 (88.7)	17,918 (90.9)	19,312 (91.1)
Education level (*n*, %)						
Primary school or below	2678 (33.9)	4311 (47.8) *	6600 (55.8)	9449 (77.6) *	9278 (47.1)	13,760 (64.9) *
Junior high school	2824 (35.8)	2612 (29.0)	3793 (32.1)	2195(18.0)	6617 (33.6)	4807 (22.7)
Senior high school or above	2387 (30.3)	2089 (23.2)	1431 (12.1)	540(4.4)	3818 (19.4)	2629 (12.4)
Marital status (*n*, %)						
Having a partner	7467 (94.7)	8506 (94.4) *	10,451 (88.4)	10,806 (88.7) *	17,918 (90.9)	19,312 (91.1) *
Other status ^	422 (5.3)	506 (5.6)	1373 (11.6)	1378 (11.3)	1795 (9.1)	1884 (8.9)
Yearly average income ^&^ (*n*, %)						
<5000 Yuan RMB	41 (0.5)	37 (0.4)	103 (0.9)	85 (0.7)	144 (0.7)	122 (0.6)
5000–9999 Yuan RMB	75 (1.0)	89 (1.0)	135 (1.1)	125 (1.0)	210 (1.1)	214 (1.0)
≥ 10,000 Yuan RMB	4096 (51.9)	4809 (53.4)	1596 (13.5)	1745 (14.3)	5692 (28.9)	6554 (30.9)
No response	3677 (46.6)	4077 (45.2)	9990 (84.5)	10,229 (84.0)	13,667 (69.3)	14,306 (67.5)
Family history of chronic diseases (*n*, %)	3014 (38.2)	3736 (41.5) *	3248 (27.5)	3499 (28.7) *	6262 (31.8)	7235 (34.1) *
Smoking ^&&^	3828 (48.5)	253 (2.8) *	6485 (54.8)	463 (3.8) *	10,313 (52.3)	716 (3.4) *
Drinking ^##^	2769 (35.1)	496 (5.5) *	4433 (37.5)	618 (5.1) *	7202 (36.5)	1114 (5.3) *
Adequate physical activity ^#^ (*n*, %)	4753 (60.2)	6547 (72.6) *	8286 (70.1)	9290 (76.2) *	13,039 (66.1)	15,837 (74.7) *
BMI (mean, std, kg/m^2^)	24.7 (3.1)	25.0 (3.2) *	23.7 (3.1)	24.4 (3.3) *	24.1 (3.1)	24.7 (3.3) *
Underweight (*n*, %)	135 (1.7)	112 (1.2) *	322 (2.7)	219 (1.8) *	457 (2.3)	331 (1.6) *
Normal weight (*n*, %)	3185 (40.4)	3493 (38.8)	6343 (53.6)	5583 (45.8)	9528 (48.3)	9076 (42.8)
Overweight and Obesity (*n*, %)	4569 (57.9)	5407 (60.0)	5159 (43.6)	6382 (52.4)	9728 (49.4)	11,789 (55.6)

^ Other marital status included those with unmarried, divorced and widowed. ^&^: Yearly average income per capital. ^&&^: Smoking once or more times during the last 30 days. ^##^: Drinking once or more times during the last year. ^#^: Moderate or above physical activity more than 150 min per typical week. * difference between gender, *p* < 0.001.

**Table 2 nutrients-14-05301-t002:** Indicators of MetS and its components prevalence among the participants aged 45 y and above in the survey.

	Urban Area			Rural Area			Total		
	Males	Females	Subtotal	Males	Females	Subtotal	Males	Females	Total
WC (mean, 95%CI, cm)	86.8 (86.3,87.3)	83.4 (82.9, 83.9)	85.0 (84.6, 85.5)	83.1 (82.5, 83.6)	81.9 (81.3, 82.5)	82.5 (82.0, 83.0)	84.8 (84.2,85.3)	82.6 (82.2, 83.0)	83.7 (83.2, 84.1)
SBP (mean, 95%CI, mmHg)	137.1 (135.9, 138.3)	138.1 (136.8, 139.4)	137.6 (136.5, 138.8)	138.5 (137.7, 139.4)	140.5 (139.6, 141.4)	139.5 (138.7, 140.3)	137.9 (137.2, 138.7)	139.4 (138.5, 140.2)	138.6 (137.9, 139.4)
DBP (mean, 95%CI, mmHg)	81.7 (81.0, 82.4)	78.0 (77.4, 78.7)	79.8 (79.2, 80.4)	81.8 (81.1, 82.4)	79.5 (79.0, 80.1)	80.7 (80.2, 81.2)	81.7 (81.2, 82.3)	78.8 (78.4, 79.3)	80.3 (79.9, 80.7)
TG (mean, 95%CI, mmol/L)	1.55 (1.51, 1.59)	1.56 (1.51, 1.60)	1.55 (1.52, 1.58)	1.43 (1.40, 1.47)	1.48 (1.44, 1.51)	1.46 (1.42, 1.49)	1.48 (1.45, 1.52)	1.52 (1.49, 1.55)	1.50 (1.47, 1.53)
HDL-C (mean, 95%CI, mmol/L)	1.20 (1.18, 1.23)	1.30 (1.28, 1.31)	1.25 (1.24, 1.27)	1.28 (1.26, 1.30)	1.32 (1.31, 1.34)	1.30 (1.28, 1.32)	1.25 (1.23, 1.26)	1.31 (1.30, 1.32)	1.28 (1.26, 1.29)
FBG (mean, 95%CI, mmol/L)	5.6 (5.52, 5.68)	5.57 (5.49, 5.65)	5.59 (5.51, 5.66)	5.45 (5.39, 5.52)	5.41 (5.35, 5.47)	5.43 (5.38, 5.49)	5.52 (5.46, 5.58)	5.49 (5.43, 5.54)	5.50 (5.45, 5.56)
Central obesity (%, 95%CI)	37.1 (34.6, 39.5)	64.7 (62.2, 67.4)	51.3 (49.1, 53.5)	24.3 (22.2, 26.4)	57.1 (54.3, 60.0)	40.2 (38.0, 42.5)	30.1 (28.0, 32.1)	60.8 (58.6, 62.9)	45.4 (43.4, 47.4)
Elevated TG (%, 95%CI)	31.8 (29.8, 33.8)	32.0 (29.9, 34.1)	31.9 (30.3, 33.5)	26.5 (25.0, 28.1)	28.2 (26.3, 30.0)	27.3 (25.8, 28.8)	28.9 (27.6, 30.2)	30.0 (28.5, 31.5)	29.4 (28.2, 30.7)
Decreased HDL-C (%, 95%CI)	30.6 (28.0, 33.2)	55.1 (52.8, 57.4)	43.2 (40.8, 45.5)	23.1 (21.2, 24.9)	50.4 (47.9, 52.9)	36.4 (34.4, 38.3)	26.5 (24.7, 28.2)	52.7 (50.8, 54.6)	39.5 (37.8, 41.3)
Elevated BP (%, 95%CI)	68.4 (65.5, 71.3)	68.0 (65.9, 70.2)	68.2 (66.0, 70.5)	68.8 (66.7, 71.0)	68.3 (66.4, 70.2)	68.6 (66.8, 70.3)	68.6 (66.8, 70.4)	68.2 (66.6, 69.7)	68.4 (66.9, 69.9)
Elevated glucose (%, 95%CI)	37.1 (34.0, 40.2)	35.6 (32.5, 38.7)	36.3 (33.5, 39.2)	32.0 (29.6, 34.4)	29.2 (27.0, 31.4)	30.7 (28.6, 32.8)	34.3 (32.2, 36.4)	32.3 (30.2, 34.4)	33.3 (31.4, 35.3)
MetS (%, 95%CI)	34.8 (32.5, 37.2)	51.4 (48.6, 54.2)	43.4 (41.1, 45.6)	25.1 (23.6, 26.6)	43.9 (41.8, 45.9)	34.2 (32.6, 35.8)	29.5 (27.9, 31.2)	47.5 (45.5, 49.5)	37.1 (35.4, 38.8)
1 items (%, 95%CI)	25.1 (23.1, 27.0)	16.9 (15.4, 18.4)	20.9 (19.4, 22.3)	33.0 (31.3, 34.6)	21.1 (19.6, 22.6)	27.2 (25.9, 28.4)	29.4 (27.9, 30.9)	19.1 (17.9, 20.2)	24.2 (23.1, 25.4)
2 items (%, 95%CI)	28.0 (26.3, 29.7)	24.6 (22.9, 26.3)	26.2 (25.2, 27.3)	27.2 (26.0, 28.4)	26.8 (25.4, 28.1)	27.0 (26.0, 27.9)	27.6 (26.5, 28.6)	25.7 (24.6, 26.8)	26.6 (25.9, 27.3)
3 items (%, 95%CI)	19.8 (18.1, 21.5)	24.8 (23.4, 26.3)	22.4 (21.1, 23.6)	14.9 (13.8, 16.0)	22.5 (21.1, 23.8)	18.6 (17.7, 19.5)	17.1 (16.0, 18.2)	23.6 (22.6, 24.6)	20.4 (19.5, 21.2)
4 items (%, 95%CI)	10.8 (9.6, 11.9)	18.0 (16.6, 19.5)	14.5 (13.5, 15.5)	8.3 (7.4, 9.2)	15.8 (14.6, 17.0)	11.9 (11.0, 12.8)	9.4 (8.6, 10.2)	16.9 (15.9, 17.8)	13.1 (12.4, 13.8)
5 items (%, 95%CI)	4.3 (3.5, 5.1)	8.6 (7.2, 10.0)	6.5 (5.5, 7.4)	1.9 (1.5, 2.4)	5.6 (4.9, 6.3)	3.7 (3.2, 4.2)	3.0 (2.5, 3.5)	7.0 (6.2, 7.9)	5.0 (4.4, 5.6)

Abbreviations: WC: Waist circumference; SBP: Systolic blood pressure; DBP: Diastolic blood pressure; TG: Triglyceride; HDL-C: High-density lipoprotein cholesterol; FBG: Fasting blood glucose.

**Table 3 nutrients-14-05301-t003:** Dietary intake among participants aged 45 y and above in three dietary patterns (mean, 95%CI, g/d).

	Diversity Pattern	Northern Pattern	Southern Pattern
N (%)	4022 (9.8)	19,288 (47.2)	17,599 (43.0)
Rice and products	78.6 (71.2, 86.0)	46.7 (42.0, 51.5)	192.1 (180.0, 204.2)
Wheat and products	130.1 (118.9, 141.3)	195.3 (182.6, 208.0)	41.6 (37.7, 45.4)
Other cereals	16.5 (14.2, 18.9)	27.7 (22.7, 32.6)	2.9 (2.3, 3.4)
Tubers	51.9 (47.6,56.3)	62.8 (54.6, 71.0)	34.5 (30.4, 38.6)
Dry legume	5.2 (4.1, 6.2)	3.5 (2.8, 4.2)	3.8 (2.9, 4.7)
Legume products	11.9 (10.5, 13.3)	7.3 (6.4, 8.2)	10.9 (9.7, 12.0)
Dark colored vegetables	112.7 (102.8, 122.6)	55.7 (51.5, 59.9)	130.5 (120.2, 140.8)
Light colored vegetables	154.3 (141.3, 167.2)	115.2 (107.0, 123.3)	149.7 (138.3, 161.2)
Salted vegetables	2.8 (2.2, 3.4)	2.5 (2.0, 3.0)	5.9 (4.7, 7.0)
Fungi and algae	24.1 (20.6, 27.5)	9.0 (7.8, 10.2)	15.0 (12.6, 17.4)
Fresh fruits	171.7 (160.7, 182.7)	21.3 (18.2, 24.5)	21.3 (18.4, 24.1)
Nuts	14.8 (13.3, 16.3)	1.6 (1.3, 1.8)	2.8 (2.4, 3.1)
Pork	49.7 (44.3, 55.1)	21.6 (19.3, 23.9)	81.1 (76.0, 86.2)
Other red meats	21.0 (16.1, 26.0)	12.8 (9.9, 15.7)	5.4 (4.4, 6.4)
Red meat offal	2.6 (1.5, 3.6)	1.5 (1.2, 1.8)	3.5 (2.8, 4.1)
Poultry	14.4 (11.8, 17.0)	4.2 (3.5, 4.9)	17.6 (15.2, 20.0)
Milk and dairy products	123.2 (110.4, 136.0)	16.1 (12.4, 19.7)	4.9 (3.5, 6.3)
Eggs	40.1 (36.9, 43.2)	19.1 (17.4, 20.9)	14.9 (13.5, 16.3)
Aquatic products	30.6 (23.7, 37.5)	6.0 (4.9, 7.1)	38.2 (32.3, 44.1)
Vegetable oil	34.3 (32.4, 36.3)	36.3 (34.4, 38.2)	36.4 (33.9, 38.8)
Animal oil	1.3 (0.7, 2.0)	1.9 (1.2, 2.6)	7.9 (6.0, 9.9)
Cakes and dessert	13.9 (11.7, 16.2)	7.8 (4.0, 11.5)	3.7 (2.4, 5.0)
Sugar and starch	9.5 (7.9, 11.1)	8.8 (7.3, 10.4)	4.5 (3.6, 5.4)
Salt	7.9 (7.3, 8.5)	9.6 (9.1, 10.0)	9.1 (8.7, 9.4)

**Table 4 nutrients-14-05301-t004:** Daily energy and nutrients intake of the three patterns among middle-aged and above participants (mean, 95%CI).

	Diversity Pattern	Northern Pattern	Southern Pattern
Daily Energy Intake (kcal/day)	1953.3 (1905.3, 2001.4)	1649.9 (1608.5, 1691.3)	1853.6 (1808.4, 1898.9)
Carbohydrate (g/day)	245.9 (238.6, 253.1)	233.9 (225.2, 242.6)	221.5 (213.2, 229.9)
Carbohydrate E%	50.4 (50.1, 52.0)	56.7 (56.0, 58.1)	48.5 (47.5, 49.5)
Protein (g/day)	67.3 (65.0, 69.6)	46.5 (45.1, 48.0)	57.1 (55.2, 59.0)
Prontein E%	13.7 (13.3, 14.2)	11.3 (11.1, 11.6)	12.5 (12.1, 12.8)
Fat (g/day)	81.8 (79.0, 84.7)	61.9 (59.8, 64.0)	83.1 (80.7, 85.5)
Fat E%	37.1 (36.3, 37.9)	33.8 (32.4, 34.5)	39.9 (38.9, 40.9)
SFA (g/day)	17.2 (16.4, 17.9)	11.5 (11.0, 12.1)	18.7 (17.7, 19.6)
MUFA (g/day)	26.8 (25.5, 28.2)	21.7 (20.4, 23.1)	32.0 (30.7, 33.3)
PUFA (g/day)	24.4 (23.3, 25.5)	19.9 (18.8, 21.1)	19.7 (18.8, 20.7)
SFA E%	7.8 (7.6, 8.1)	6.3 (6.0, 6.5)	8.9 (8.6, 9.3)
MUFA E%	12.2 (11.7, 12.6)	11.7 (11.1, 12.4)	15.4 (14.8, 16.0)
PUFA E%	11.2 (10.7, 11.6)	10.8 (10.2, 11.5)	9.5 (0.1, 10.0)
Dietary fiber (g/day)	13.4 (12.9, 13.9)	9.4 (9.0, 9.7)	8.7 (8.3, 9.0)
Cholesterol (mg/day)	200.2 (178.4, 221.9)	74.0 (66.0, 81.9)	182.3 (169.8, 194.7)
Thiamin (mg/day)	0.88 (0.85, 0.91)	0.71 (0.68, 0.74)	0.73 (0.71, 0.76)
Riboflavin (mg/day)	0.96 (0.92, 1.00)	0.54 (0.52, 0.56)	0.69 (0.66, 0.71)
Vitamin A (μg RE/day)	567.6 (534.2, 601.1)	272.4 (252.3, 292.4)	467.6 (440.5, 494.7)
Vitamin C (mg/day)	109.6 (101.0, 118.1)	59.4 (56.5, 62.3)	84.2 (80.3, 88.0)
Total α-vitamin E (mg/day)	11.8 (10.9, 12.6)	7.8 (7.2, 8.3)	8.3 (7.9, 8.7)
Calcium (mg/day)	519.3 (493.3, 545.3)	270.1 (261.0, 279.3)	347.1 (330.9, 363.2)
Copper (mg/day)	2.2 (2.0, 2.3)	1.6 (1.3, 1.8)	1.5 (1.5, 1.6)
Iron (mg/day)	22.2 (21.5, 23.0)	17.6 (16.9, 18.3)	19.4 (18.7, 20.1)
Iodine (μg/day)	8.3 (5.9, 10.6)	7.0 (5.6, 8.5)	5.9 (5.1, 6.8 )
Potassium (mg/day)	2023.0 (1955.8, 2090.2)	1274.9 (1233.5, 1316.3)	1435.4 (1393.0, 1477.8)
Magnesium (mg/day)	305.9 (296.8, 314.9)	233.7 (225.9, 241.6)	234.3 (227.4, 241.3)
Manganese (mg/day)	5.1 (5.0, 5.3)	4.5 (4.3, 4.6)	5.0 (4.8, 5.1)
Sodium (mg/day)	6583.0 (6200.5, 6965.4)	7509.5 (7190.0, 7829.0)	7231.7 (6968.8, 7494.6)
Phosphate (mg/day)	1026.1 (991.6, 1060.5)	734.8 (711.3, 758.3)	806.0 (785.4, 826.5)
Selenium (μg/day)	48.8 (46.4, 51.1)	34.8 (33.4, 36.2)	38.0 (35.6, 40.5)
Zinc (mg/day)	10.7 (10.3, 11.1)	7.7 (7.2, 8.1)	9.9 (9.6, 10.2)

Abbreviations: SFA: saturated fatty acids; MUFA: mono-unsaturated fatty acids; PUFA: polyunsaturated fatty acids. E%: daily energy from the macronutrients.

## Data Availability

The data presented in this study are not allowed to disclose.

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
