# Peer review of "MetS Prevalence and Its Association with Dietary Patterns among Chinese Middle-Aged and Elderly Population: Results from a National Cross-Sectional Study"

_nutrients, 2022, doi:10.3390/nu14245301_

Round 1

Reviewer 1 Report

The article entitled "MetS prevalence and its association with dietary patterns among Chinese middle-aged and above population: results from a national cross-sectional study" aimed to investigate the prevalence of MetS and its components, the dietary patterns, and their relationship among the middle-aged and above population of China.

The study results can serve as input to focus efforts on public policies. I hope my comments serve to improve the manuscript.

Introduction:

- Is there more recent data on MetS prevalence?

- Please consolidate abbreviations

- I suggest including the most common criteria for MetS definition

- Line 55: What do the authors mean by "obviously different"?

Results:

- In the table ** means p<0.01 between rural and urban or between male and female? Is the significance p<0.01 or 0.001?

- Table 2, 3, and 4: table legend is missing

- It it not clear to me the classification in diversity pattern, northern pattern, or southern pattern. Could you explain that?

- Table 4: n-3 PUFA information would be interesting to show since those fatty acids are known to prevent MetS metabolic alterations. I recommend analyzing according to fatty acids profile and not just total fat.

Author Response

Introduction:

- Is there more recent data on MetS prevalence?

Author response: Thanks for your attention on the MetS prevalence of China. As far as I know, the published results on the prevalence of metabolic syndrome in Chinese middle-aged and elderly population was from two nationwide studies: (1) the 2015 CHERLS study showed that the prevalence of metabolic syndrome was 34.77% by using revised ATP III criteria; (2)The China National Stroke Prevention Project (CSPP) survey in 2014–2015 included 109,551 participants aged ≥40 years from 30 provinces in China, and the prevalence of MetS was 34.0% by the revised ATP III criteria. In our study, this is the most recent available data conducted in 31 provinces in China mainland as far as I know, and the sampling population is both national and provinces representative.

- Please consolidate abbreviations

Author response: I will check the manuscript and consolidate abbreviations, thanks.

- I suggest including the most common criteria for MetS definition.

Author response: Thanks for your suggestion. The common criteria for MetS definition are the modified ATP III and IDF definition, and study reported that agreement between IDF and modified ATP III criteria was good (Kappa = 0.83). Because the cut-off of five components of MetS in IDF definition was the same as that in modified ATP III, and the two definition was common used in Asia population, the modified ATP III might be more suitable for the Chinese population to detect and treatment MetS earlier. We checked the analysis procedure carefully, and the modified ATP III criteria was used to calculate the prevalence of MetS in the present manuscript. It was our default in the description in the method. I will revised it carefully. The results using IDF definition was also showed in the table 2.   

- Line 55: What do the authors mean by "obviously different"?

 Author response: I make a modification of this sentence. Please see the part of introduction.

Results:

- In the table ** means p<0.01 between rural and urban or between male and female? Is the significance p<0.01 or 0.001?

 Author response: The means or proportion tested in the table were between male and female. Thanks for your reminder, I checked the p value and make revision in the footnote, the significance was p<0.001.  

- Table 2, 3, and 4: table legend is missing.

 Author response: I added the table legend, thanks.

- It is not clear to me the classification in diversity pattern, northern pattern, or southern pattern. Could you explain that?

 Author response: Thanks for your question. I have added description in the method of extracting dietary pattern procedure.

- Table 4: n-3 PUFA information would be interesting to show since those fatty acids are known to prevent MetS metabolic alterations. I recommend analyzing according to fatty acids profile and not just total fat.

 Author response: Thanks for your suggestion, I have added results of fatty acids profile including daily consumption of SFA, MUFA, and PUFA as well as the percentage of total energy that comes from the three fatty acids.  

Reviewer 2 Report

The authors aimed to investigate the prevalence of Metabolic syndrome and its association with the dietary patterns among the middle-aged and above population of China. Data were obtained from a national cross-sectional survey named „China Adults Chronic Diseases and Nutrition Surveillance“ (CACDNS 2015) conducted in 2015. This study was conducted in 31provinces, autonomous regions, and municipalities throughout China. Among 40,909 middle-aged and older participants, the prevalence of Metabolic syndrome was 37.1%. The study revealed that the three dietary district-specific patterns have their own nutritional characteristics and need to be improved in order to prevent and control Metabolic syndrome and improve quality of life. This manuscript reviews 40 articles. The topic of this manuscript is up-to-date, attractive and well-suited for the Journal Nutrients. The manuscript is well-written and divided into five parts. The authors included 4 tables and 1 figure. The text is easy to read, but some parts require minor modifications - checking for some minor spelling mistakes and grammar errors, maybe a change of word order. Otherwise, I  have no major concerns regarding this manuscript.

Author Response

Comments and Suggestions for Authors

The authors aimed to investigate the prevalence of Metabolic syndrome and its association with the dietary patterns among the middle-aged and above population of China. Data were obtained from a national cross-sectional survey named „China Adults Chronic Diseases and Nutrition Surveillance“ (CACDNS 2015) conducted in 2015. This study was conducted in 31provinces, autonomous regions, and municipalities throughout China. Among 40,909 middle-aged and older participants, the prevalence of Metabolic syndrome was 37.1%. The study revealed that the three dietary district-specific patterns have their own nutritional characteristics and need to be improved in order to prevent and control Metabolic syndrome and improve quality of life. This manuscript reviews 40 articles. The topic of this manuscript is up-to-date, attractive and well-suited for the Journal Nutrients. The manuscript is well-written and divided into five parts. The authors included 4 tables and 1 figure. The text is easy to read, but some parts require minor modifications - checking for some minor spelling mistakes and grammar errors, maybe a change of word order. Otherwise, I have no major concerns regarding this manuscript.

Author response: Thanks for your comments, I will check the manuscript carefully not only the spelling and grammar mistakes but also the results description.